# Simulation and Experimental Study on the Internal Leak Behavior in Carbon Fiber Reinforced Composite Components

**DOI:** 10.3390/polym15132758

**Published:** 2023-06-21

**Authors:** Shu Liu, Lihua Zhan, Bolin Ma, Chenglong Guan, Xiaobo Yang

**Affiliations:** 1State Key Laboratory of Precision Manufacturing for Extreme Service Performance, Central South University, Changsha 410083, China; ls88god@163.com (S.L.); yjs-cast@csu.edu.cn (L.Z.); 2School of Mechanical Engineering and Automation, Fuzhou University, Fuzhou 350108, China; guancl@fzu.edu.cn; 3School of Mechanical Engineering, Lanzhou Jiaotong University, Lanzhou 730070, China; lzjt_yangxb@163.com

**Keywords:** carbon fiber reinforced composite, leak rate, finite element simulation, leak experiment

## Abstract

In this paper, the diffusion law of helium gas inside composite materials was obtained through numerical research and an experimental approach. The influence of fiber and the fiber–resin interface on permeability was discussed in the actual numerical model. It was found that the leak rate and the mass concentration at the fiber–resin interface were higher than those in the resin, and the leak rate symmetrically distributed along the horizontal central line. Meanwhile, a homogenized model for the leak rate simulation in carbon fiber composite components was established, and its accuracy was verified through the experiment and the actual numerical model. The simulated result and the test data demonstrated that the leak rate increased with the pressure and decreased with the thickness of the specimen.

## 1. Introduction

The aerospace manufacturing industry is one of the rapidly developing and most leading strategic industries in the world [1]. Many countries have comprehensively deployed major strategic tasks such as manned lunar exploration and deep space exploration, and the demand for carrying capacity has developed from the ten-ton level to the hundred-ton level [2,3]. The high-quality manufacturing of lightweight materials and their components is the key to achieving this point. At the same time, high-density launch missions urgently need the development of spacecraft manufacturing capabilities towards efficient, low-cost, and batch production, which challenges the formability of key load-bearing parts of spacecraft storage tanks. Compared with metal storage tanks, carbon fiber reinforced resin-based composite storage tanks can achieve a weight reduction of 20–40% and a comprehensive cost reduction of more than 25% [4,5,6]. Therefore, large-scale composite storage tanks are the key to improving carrying efficiency and reducing costs [7]. Currently, large-scale composite storage tanks face problems such as the difficulty in ensuring high forming accuracy, chemical compatibility issues, the occurrence of microcracks during service, and the leak of small-molecule propellants at a low temperature [8,9,10]. Among them, the leak of small-molecule propellants at a low temperature is the main factor restricting the use of large-scale composite storage tanks [11,12]. When the gas leak reaches a certain level, it would induce structural failure or gas explosion accidents [13]. Therefore, many researchers have carried out experimental, theoretical, and numerical research on the leak phenomenon.

In the field of experimental research, the works presented by M. Flanagan indicated that the leak in the composite components mainly occurs in the form of molecular diffusion when the composite parts with good curing quality are not subject to load impact or structural damage [6]. Stokes studied the leak rate of composite flat plates under the condition of four-axis loads, and the evolution of the small cracks and the leak in composite components was presented [14]. Meanwhile, NASA studied the failure of composite material tanks in the SPACE X spacecraft; it was found that small cracks in the composite components expand under the temperature and force loads, and eventually, the leak path that penetrates the top and bottom surfaces of the component forms [15]. Based on this existing literature, it can be found that the main forms of leak in the composite components are the diffusion leak of media at low-temperature and the penetration leak in the resin matrix crack. Diffusion leak is caused by the Brownian motion of gas molecules. Because of the concentration gradients along the part thickness direction, the propellant molecules inside the tank diffuse outward along the composite tank wall, and it results in the gas leak. Currently, many experimental works have been carried out to study the leak phenomena in composite parts, and theoretical research is further needed to reveal the mechanisms behind these experimental results.

Regarding the theoretical research on gas leaks, Riyuan Xia et al. established a diffusion and surface release model to describe the leaking of inert gas in metal materials through considering defect and radiation damage, and the proposed diffusion equations were analytical solved [16]. A mathematical model for expressing the diffusion of water molecules in composite materials was proposed by Lee et al. through considering the factor of stress, and the proposed model under appropriate boundary conditions can well solve the concentration and stress distribution [17]. The results showed that the diffusion of water molecules in composite materials followed Fick’s law, and the tensile stress of the component was positively correlated with the diffusion rate of water molecules. Shunying Tang et al. found that the diffusion of organic low-molecular-weight compounds in polymers is determined by the relative motion between the permeant molecules and the polymer chain segments, so the leak rate is affected by factors such as temperature, permeant concentration, molecular size and shape, and polymer structure [18]. It can be found from the existing works that the theoretical model for the leak is mature, but its applicability is poor in the leak rate evaluation procedure of the composite components with a large size or complex structure. The current approach for the above leak problem is to solve the complex theoretical model through numerical simulation.

In the leak simulation work, a diffusion model was established by Sun Li et al. through using ABAQUS software to study the leak of water molecules in carbon fiber reinforced resin-based composite materials; they found that the temperature was positively correlated with the diffusion rate of water molecules [19]. Guan et al. established a simulation model of a random porous leak for carbon fiber composite materials through gray processing, median filtering, and random medium theory [20]. The research results showed that the leak rate of composite component increases with the porosity. In the case of the same porosity, the leak rate of laminates with smaller average pore diameter is higher than that of laminates with larger average pore diameter. These works focus on the effects of factors such as porosity and temperature on the leak of composite materials; they simplified the composite material as a resin matrix or only considered the fiber and fiber–resin interface at a small scale, and the effect of the fibers and the resin–fiber interfaces on the leak was ignored. Hence, it would lead to a significant difference between the simulation and the actual situation when the current simulation approach is applied for evaluating a large-scale leak. Meanwhile, if a large-scale leak model considering the actual structure is established, the huge number of elements in the model leads to a decrease in computational efficiency.

In this paper, a theoretical leak prediction model was proposed, and based on this leak theory, a leak simulation model considering the resin–fiber interface and the resin was established under the quasi-static condition by using ABAQUS software. The simulation results were discussed, and it was found that the leak rate at the fiber–resin interface is higher than that in the resin, and the helium gas flow rate in the interface of composite components is horizontally symmetrically distributed. After that, a method of homogenizing composite structure was proposed, and a full-size homogenized model for simulating the leak of composite component was established. Through comparing the test data with the simulation results determined by the full-size homogenized model, the feasibility of homogenization method was proved. Furthermore, the experiment data and the simulation results showed that the leak rate is directly proportional to the medium pressure and inversely correlated with the component thickness.

## 2. Leak Prediction Model

### 2.1. Diffusion Leak Principle of Composite Components

For the composite component under the quasi-static condition, the main leak way is the diffusion leak. It can be divided into four stages: adsorption, dissolution, diffusion, and desorption. First, the composite component surface facing the high pressure and high medium concentration adsorbs the gas molecules. Then, gas molecules enter the inner layer of the component until a certain medium concentration corresponding to the environmental pressure is reached. Subsequently, gas molecules move towards the other side with low pressure and low concentration because of the concentration gradient, and subsequently, they are desorbed. The generalized chemical potential control equation can well describe the helium gas leak procedure, and it can be expressed as Equation (1):(1)$$J\;=\;-\;S\;D\;[\frac{\partial \Phi }{\partial x}+\kappa_{s}\frac{\partial }{\partial x}\left(\ln\left(\theta -\theta_{z}\right)\right)+\kappa_{P}\frac{\partial P}{\partial x}]$$
where symbol *J* is the concentration flux of the diffusing substance, and *S* and *D* indicate the solubility of the substance and the diffusion coefficient, respectively. Parameter *Φ* represents the normalized concentration, and it can be regarded as the driving force for diffusion, defined as *Φ* = *C*/*S,* where *C* means the concentration of the substance. The Soret effect factor *κ_s_* is used to control temperature-driven mass diffusion and the pressure factor *κ_P_* means the pressure-driven mass diffusion. Symbols *θ* and *θ_z_* are the temperature and the absolute zero temperature, respectively. Parameter *P* is the equivalent pressure.

Since the diffusion leak behavior of composite components is driven by the mass concentration, and the temperature is constant during the leak process, the mass diffusion driven by temperature can be neglected, and $\frac{\partial }{\partial x}\left(ln\left(\theta -\theta_{Z}\right)\right)$ is equal to 0. Therefore, Equation (1) can be transformed as follows:(2)$$J\;=\;-\;S\;D\;(\frac{\partial \Phi }{\partial x}+\kappa_{P}\frac{\partial P}{\partial x})\;$$

Considering the relationship of *Φ* = *C*/*S*, Equation (2) can be written as:(3)$$J=-D\;(\frac{\partial C}{\partial x}\;+S\kappa_{P}\frac{\partial P}{\partial x})$$

With the ideal gas equation *PV* = *n⋅RT* and the molar concentration definition c = n/V, the relationship among the pressure *P* and the temperature *T* is determined as Equation (4) through taking the value of *R* as 8.314 J/(mol·K):*P* = 8.314*c⋅T*
(4)

Setting the pressure stress factor *κ_p_* as 1 and substituting Equation (4) into Equation (2), the final mass diffusion control equation under the specific temperature is expressed as:(5)$$J=-D\;(1+8.314\mathrm{ST})\;\frac{\partial C}{\partial x}$$

### 2.2. Actual Diffusion Leak Numerical Model of Composite Structural Components

#### 2.2.1. Model Construction

An actual structural was established in the simulation process based on the actual size of the microstructure of the composite component; the detail size in the model is shown in Figure 1. The upper blue part represents high-pressure helium gas, the lower blue part means vacuum, the green parts are the resin, and the yellow part and the black part indicate the fiber–resin interface and the carbon fiber, respectively.

In this numerical model, the predefined temperature field was set to 77 K, and an 8-node linear thermal conductivity element DC3D8 was selected. The mass diffusion module and two analysis steps were applied to analyze the diffusion leak process. The first analysis step typed as steady state was used to set the helium concentration on one side of the laminate corresponding to the experimental data, and the second analysis step typed as transient was used to study the diffusion of helium gas inside the composite material laminate and pores. For the boundary conditions in this numerical model, the gas concentration on the surface remains constant during the leak process. The values of the solubility *S* and the diffusivity *D* in this numerical model were determined experimentally, and they are shown in Table 1 [8].

In order to clear the effect of the mesh size on the simulation result, the model was meshed by using mesh sizes of 0.001 mm, 0.0005 mm, and 0.0001 mm, respectively. The simulated results are shown in Figure 2.

In Figure 2a, the simulated maximum helium concentration changes from 5.117 × 10^−7^ to 5.120 × 10^−7^ when the mesh size decreases from 0.001 mm to 0.0001 mm; this slight difference among the leak rates can also be confirmed in Figure 2b. Meanwhile, Figure 2 shows that the simulated mass concentration distributions or the leak rate distributions under the different mesh sizes are the same. Therefore, it can be confirmed that the effect of the mesh size on the simulation result is slight.

#### 2.2.2. Simulation Results

Based on the previous mesh sensitivity study and considering the computational efficiency, the mesh size at the resin and the fiber–resin interface was set to 0.0002 mm, while it was 0.0005 mm at the fibers. The hexahedral element is generated through applying the sweeping method, and the element number in this model is 39,988. In order to obtain the helium gas leak rate and the gas concentration from the simulation result, 10 points marked as F1–F5 and P1–P5 were selected, where F1–F5 is located at the fiber–resin interface and P1–P5 are in the resin. The positions of these points are shown in Figure 3. The simulated data at each point was organized and plotted in Figure 4.

Figure 4a,c shows that the helium gas concentration in both the resin and the interface increases sharply at the leak beginning. When it reaches the steady stage, the concentration values at P1–P5 are significantly different, while those at F1–F5 are very close. Meanwhile, it can be found that the gas concentration at the fiber–resin interface is about 5 × 10^−4^ kg/m^3^; it is much higher than that at the resin region; the fiber–resin interface is the main leak channel. The concentration gradient difference from point F1 to point F5 is not significant, while for point P1 to P5, the concentration gradient difference is obvious. The concentration values of P1–P5 at the steady state are consistent with the data shown in Figure 5a. Furthermore, Figure 4b,d show that the leak rate at the steady state is symmetrically distributed along the horizontal central line. At the fiber–resin interface (shown in Figure 4d), the leak rate at each point in the steady state decreases with the vertical distance between its position and the point and the horizontal central line, and the maximum leak rate appears at the F3 point, while the leak rate distribution of the points in the resin is opposite to that of points at the interface, and the maximum leak rate appears near the component surface (P1 or P5 shown in Figure 4b).

In this simulation, the molecule diffuses from the top to the bottom surface, the fibers are non-diffusion phase, and the diffusion coefficient and the solubility in fiber part are both set to zero. Therefore, the width of the diffusion path first decreases and then increases. According to the formula of leak amount *Q* = *S*·●*V* (where *S* and *V* are the flow cross-sectional area and the leak rate, respectively), the leak rate increases with the decrease of the flow cross-sectional area when the leak amount *Q* is constant. Therefore, the flow cross-section of helium gas leak rate in the fiber–resin interface is largest at the horizontal axis and the smallest value appears at the upper or lower surfaces.

However, the leak rate distribution in the resin part is opposite to that in the fiber–resin interface, i.e., the leak rate at the point increases with the distance between its position and the horizontal axis. Since the leak rate of helium gas in the fiber–resin interface is higher than that in the resin phase, according to the Bernoulli’s principle, where the leak rate is high, the pressure is low. Therefore, the pressure in the fiber–resin interface is lower than that in the resin, and the helium gas in the resin phase flows toward the interface. Based on the flow velocity law in the interface phase, the pressure in the interface increased with its vertical distance to the center axis, and therefore, the simulated leak rate distribution in the resin part is determined.

In addition, the simulated results at the points of P1 and P2 in Figure 4b and the points of F1 and F2 in Figure 4d show a phenomenon of reaching peak values first and then falling back to steady-state values. This is due to the mass scaling function in the model; it causes excessive inertia force at the simulation beginning and results in distorted transient response results. The inertia force stabilizes over time, and consequently, the transient response results fall back to steady-state values.

Furthermore, the leak rate and the concentration at P3 and F3 points were extracted. The leak rate and the concentration between the two points were plotted and shown in the following Figures. Although the vertical distance of each point is the same, and the helium gas leak and the mass concentration at each point are different, the leak rate and the mass concentration at the fiber–resin interface are much higher than those at the resin part, and the fiber–resin interface is the main diffusion channel.

Figure 6 shows the gas leak rate distribution and the mass concentration distribution at the steady state, and three paths marked as PAHT1, PATH2, and PATH3 were marked. As shown in Figure 6a, the gas leak rate at the fiber–resin interface is larger than that in the resin, and the maximum leak rate appears at the horizontal central line. In Figure 6b, the simulated mass concentration at the fiber–resin interfaces is the largest and distributed uniformly, while in the resin, the mass concentration decreases along PATH2. In summary, the leak rate and the mass concentration at the fiber–resin interface are higher than those at the resin, and it can be confirmed that the fiber–resin interface is the main leak channel.

In order to show the simulation result, the leak rate and the mass concentration along each path were extracted and plotted in Figure 7. As shown in Figure 7a, the distribution pattern of the helium leak rate along PATH1 is similar to that of the mass concentration at the escape interface. The leak rate or mass concentration increases near the fiber–resin interface and decreases in the resin. After integrating, the total helium leak rate in the actual structural model was about 8.05 × 10^−16^ kg/s. For PAHT2 and PATH3, the distributions of the leak rate and the mass concentration are shown in Figure 7b,c, respectively.

As shown in Figure 7b,c, the helium gas leak rate distributed symmetrically along the horizontal central line, the mass concentration decreases along the PATH2 direction. This simulated relationship among the leak rate and the mass concentration can be explained by Formula (5), the leak rate is in proportion to the derivative of concentration with respect to displacement. In Figure 7c, a sharp increase of the mass concentration and the leak rate can be observed when the path passes through the fiber–resin interface, it can be attributed to the significant difference in the helium gas leak rate and the concentration of the resin and the fiber-resin interface. Meanwhile, the leak rate under different helium pressure was investigated through this numerical approach through adjusting the pressure load, and the results are shown in Figure 8. A linear relationship among the helium pressure and the component leak rate can be observed.

### 2.3. Homogenized Model of Composite Material Component

#### 2.3.1. Model Construction and Parameter Calculation

Based on the dimensions in the previous actual model, a homogenized numerical model was established, and shown in Figure 9. The upper blue part represents high-pressure helium gas, the lower blue part means vacuum, and the middle orange part represents the homogeneous material. The boundaries in this homogenized model are the same as with the set in the previous model.

In the homogenized model, the diffusion coefficient *D*_0_ and solubility *S*_0_ are constant, and a corresponding leak rate *J*_0_ can be simulated; the relationship among the *J*_0_ and *D*_0_* *S*_0_ can be expressed in Equation (6) through introducing parameter *θ_KJ_*. It is obvious that the value of *J*_0_ based on the homogenized model is different with the actual leak rate *J*, and the values of *D*_0_ and *S*_0_ can be corrected by the difference between *J* and *J*_0_:(6)$$\theta_{KJ}=\;\frac{J_{0}}{S_{0}\times D_{0}}$$
where the unit of parameter *θ_KJ_* is m^−1.^

First, the values of parameter *S*_0_ and *D*_0_ in the homogenized model are set to be any reasonable value, such as *S*_0_ = 1 × 10^−9^ kg/m^3^ and *D*_0_ = 1 × 10^−8^ m^2^/s, and consequently, the corresponding leak rate *J*_0_ is calculated as 6.9425 × 10^−13^ kg/(m^2^s). Through applying Equation (6), the value of the correction factor *θ_KJ_* is determined as 69,425/m.

Next, the simulated leak rate *J* based the actual structure model is used, and the values of *S*_0_ and *D*_0_ are corrected. The leak rate *J* in the actual model is 3.03325 × 10^−14^ kg/(m^2^s). The permeability coefficient *S*_1_**D*_1_ in the homogenized model is calculated as: $\frac{J}{\theta_{KJ}}$ = 4.369 × 10^−19^ kg/(m·s), where *S*_1_ and *D*_1_ are the corrected solubility and the corrected diffusion coefficient, respectively.

Based on the value of *S*_1_**D*_1_, a reasonable value, such as *S*_1_ = 1 × 10^−10^ kg/m^3^, is selected, and the corrected diffusion coefficient *D*_1_ in the homogenized material can be determined as 4.369 × 10^−9^ (m^2^/s). After that, the corrected values of *S*_1_ and *D*_1_ can be used in the homogenized model.

Based on the above description, the corrected simulated leak rate *J*_1_ from the homogenized model is 3.03318 × 10^−14^ kg/(m^2^·s); it is very close to the result from the actual structure model (3.03325 × 10^−14^ kg/(m^2^·s)). Meanwhile, these corrected parameters are applied in the following simulation.

#### 2.3.2. Applicability of Homogenized Model

In order to further verify this homogenized model, the models with 1–8 basic units in the thickness direction and the width direction were developed, respectively. The pressure load was set to 0.1 MPa, and the simulated results based on the actual model are shown in Figure 10.

The overall leak rate for each model was obtained by integrating the leak rate at the bottom of the model, as shown in Figure 11. For the models considering the thickness, the maximum difference in leak rate among the previous model and the homogenized model is 2.62 × 10^−21^ kg/s, while as the width is introduced, the maximum difference between each simulated result is 1.21 × 10^−20^ kg/s. Through comparison with the simulated leak rate from the previous model, it can be found that the prediction error of the homogenized model is less than 2%.

Meanwhile, the homogenized model under the different pressure was numerically solved, and the prediction result was compared with that of the previous model. For each model, the layer number is 11, and the pressure is listed as 0.1, 0.2, 0.3, 0.4, and 0.5 MPa. The simulated leak rate from each model under different pressure conditions is compared in Figure 12. It can be found that the perdition difference among each model is slight, and furthermore, the leak rate increases with the pressure.

## 3. Experiments

### 3.1. Materials and Method

The material used in this study is carbon fiber (T800)/epoxy prepreg (Aerospace Long March Arimt Technology Co., Ltd, Tianjin, China), with a fiber volume fraction of 67% and a density of 1.25 g/cm^3^. The ply thickness is 0.125 mm. In order to match the simulation model and prevent the laminate from cracking during the leak test procedure, a unidirectional laminate with a thickness of 10 plies and a size of 150 × 150 mm are used for the ply layup.

The laminated plate was cured according to the standard curing process, then cut into circular specimens with a diameter of 100 mm; the specimen edge was polished. The standard curing process curve for the material is shown in Figure 13. 

### 3.2. Leak Test

An in-house leak test system at low temperature was used in this study, as shown in Figure 14. The leak test system mainly consists of high-pressure helium cylinders, helium mass spectrometer leak detectors (Hua Er Sheng Intelligent Control Technology Co., Ltd., Shenzhen, China), upper and lower leak testing chambers, liquid nitrogen cryogenic boxes (HM, Shanghai, China), pressure sensors, liquid nitrogen tanks, and so on [21]. The helium mass spectrometer leak detector is used to measure the leak rate, and the accuracy is 1 × 10^−15^ Pa·m^3^/s.

The leak test procedure is described as follows:

(1) The test piece F was fixed in the leak test chamber DE, and then the pressure in the chamber was increased until it reached the set value. The pressure during the test was measured and collected by the pressure gauge G.

(2) After keeping the pressure for four hours, the pressure variation was recorded for checking the air tightness of the chamber, and the air pressure change should be within 1%. After that, the temperature and the time of the cryogenic box were set.

(3) When the temperature of the cryogenic box reached the set value, the high-pressure helium gas was allowed to enter the upper leak test chamber, and the helium gas pressure was adjusted to the test standard. Then, the helium gas leak rate was measured and recorded by detector A.

### 3.3. Experimental Results and Analysis

The methods for gas storage include ambient temperature high-pressure gas storage, cryogenic liquid storage, organic liquid storage, and solid-state storage [22]. Cryogenic liquid storage is typically used for spacecraft, and the relevant data of the spacecraft tanks are confidential. Referencing the automotive liquid hydrogen tank designed by the Germany industrial gas and engineering company Linde (Shanghai, China) [23], the internal pressure of the tank was set to be 0.5 MPa in this work. Based on the homogenized numerical model and the leak experiment, the leak rate was listed in Table 2 and compared in Figure 15.

The experimental data demonstrated that the leak rate increases with the pressure; it is consistent with the simulation results. Meanwhile, it can be found that the simulated leak rate is close to the numerical data, and the error between the simulated and experimental results is within 20% in the range of 0.1 MPa to 0.5 MPa. It can be concluded that the simulation and experiment are in good agreement.

## 4. Conclusions

This paper focuses on the leak of the composite components. A leak numerical model considering the carbon fiber–resin interface and the resin was developed and homogenized, and the simulation result was discussed. Furthermore, an in-house leak test device was applied to conduct the leak test, and the test data were compared with the numerical data. The feasibility of the homogenized numerical approach was demonstrated and the following rules can be listed:

1. The helium gas leak rate and the mass concentration at the fiber–resin interface are higher than those in the resin; the fiber–resin interface is the main leak channel.

2. The leak rate is symmetrically distributed along the horizontal central line. For the resin part, the leak rate increases with the distance from the central axis, while for the fiber–resin interface, the max leak rate appears at the central axis.

3. The leak rate of the composite component is proportional to the pressure and inversely related to the thickness of the component.

## Figures and Tables

**Figure 1 polymers-15-02758-f001:**
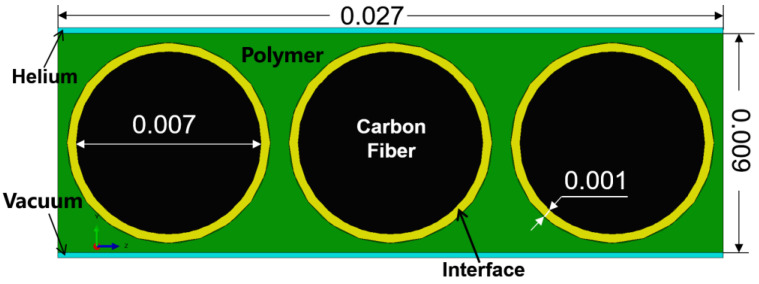
The actual structure in the numerical model (unit: mm).

**Figure 2 polymers-15-02758-f002:**
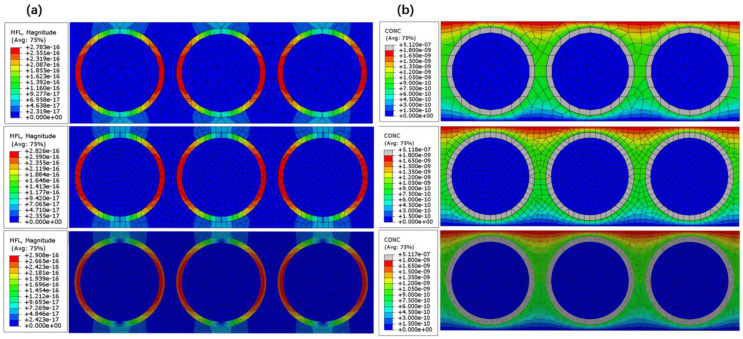
Simulated results under different mesh sizes (0.001 mm, 0.0005 mm, and 0.0001 mm): (**a**) Helium concentration, (**b**) Helium leak rate.

**Figure 3 polymers-15-02758-f003:**
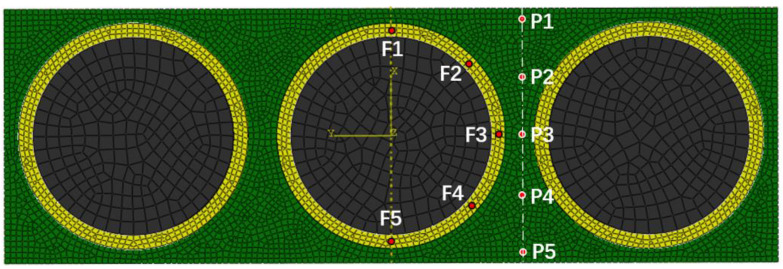
Locations of the points.

**Figure 4 polymers-15-02758-f004:**
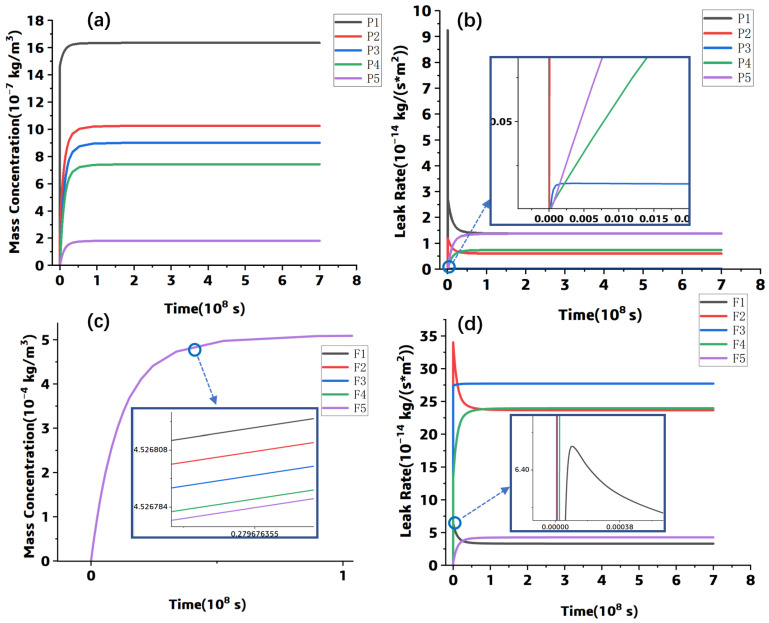
The simulation results: (**a**) concentration-time curves at P1–P5, (**b**) leak rate-time curves at P1–P5, (**c**) concentration-time curves at F1–F5, (**d**) leak rate-time curves at F1–F5.

**Figure 5 polymers-15-02758-f005:**
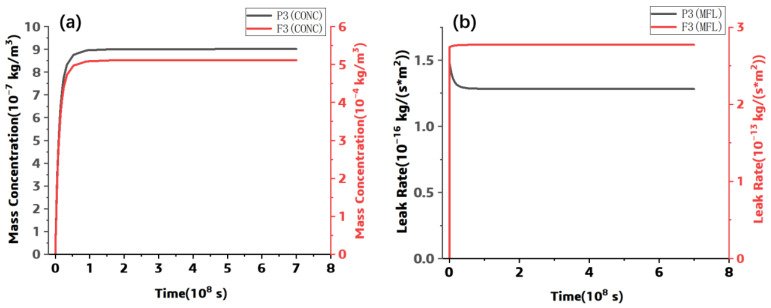
The simulated results at P3 and F3: (**a**) Concentration, (**b**) Leak Rate.

**Figure 6 polymers-15-02758-f006:**
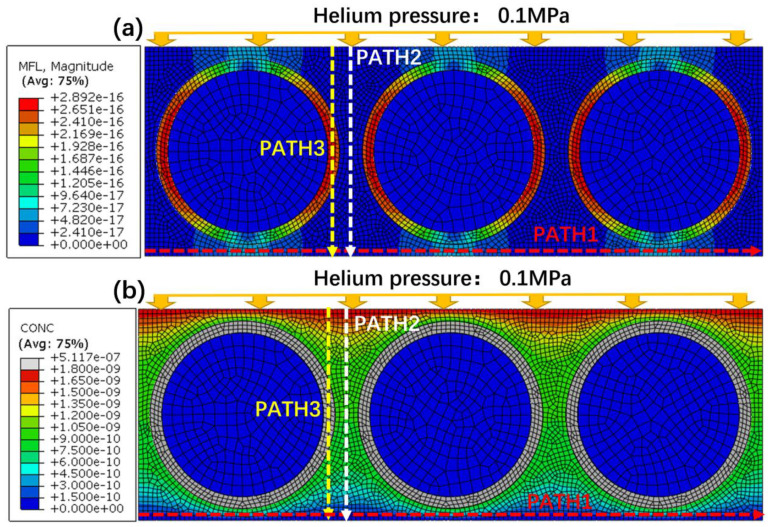
Simulated cloud diagrams of the actual structural model: (**a**) Helium leak rate (**b**) Helium concentration.

**Figure 7 polymers-15-02758-f007:**
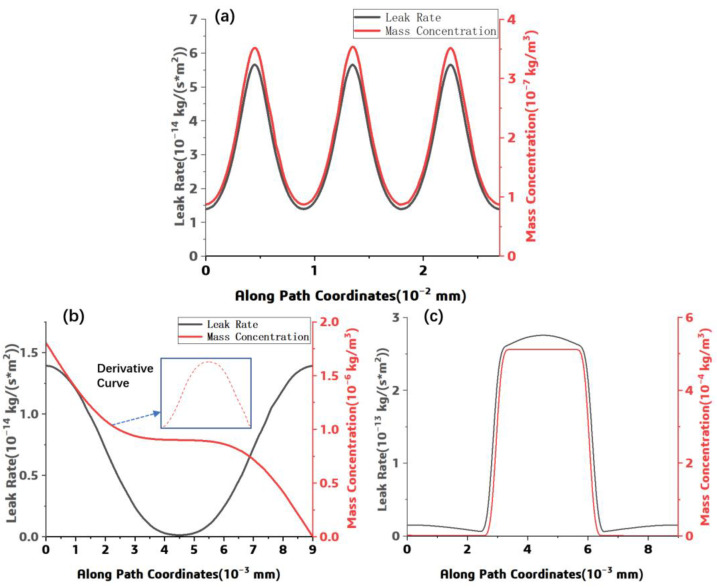
Helium gas leak rate/concentration variation function along path: (**a**) PAHT1 (**b**) PATH2, (**c**) PATH3.

**Figure 8 polymers-15-02758-f008:**
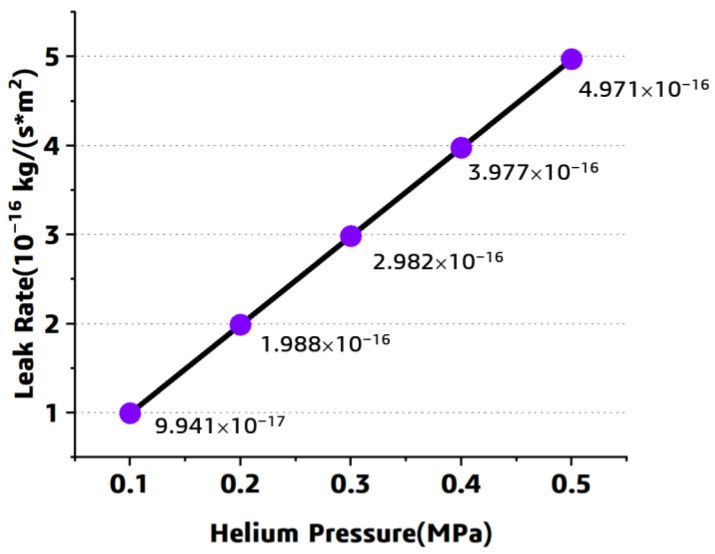
The simulated results under different pressure.

**Figure 9 polymers-15-02758-f009:**
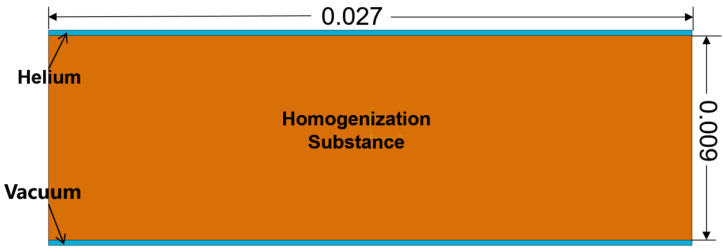
The homogenized model (unit: mm).

**Figure 10 polymers-15-02758-f010:**
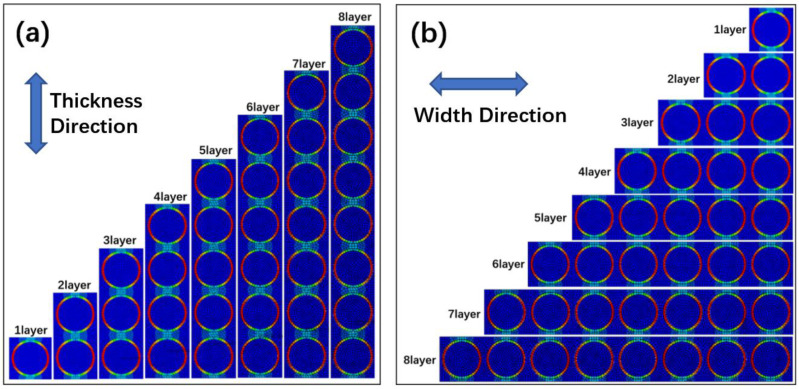
The simulated results: (**a**) different thickness (**b**) different width.

**Figure 11 polymers-15-02758-f011:**
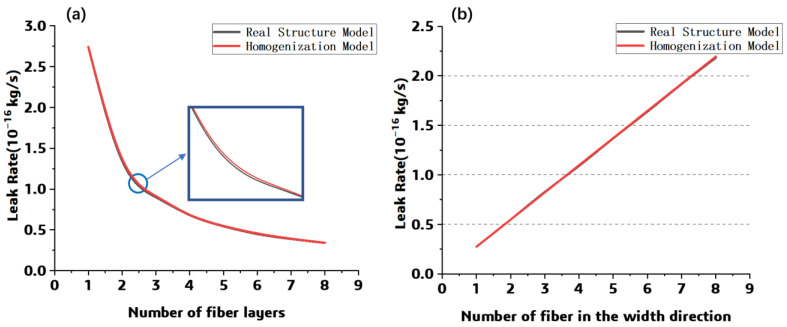
The numerical result comparison: (**a**) different thickness (**b**) different width.

**Figure 12 polymers-15-02758-f012:**
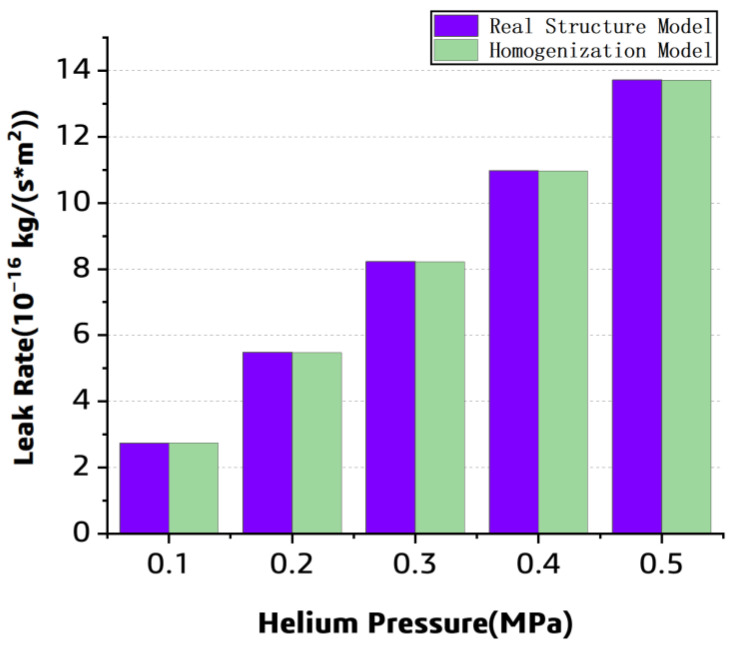
The simulated helium leak rate pressure from the two models.

**Figure 13 polymers-15-02758-f013:**
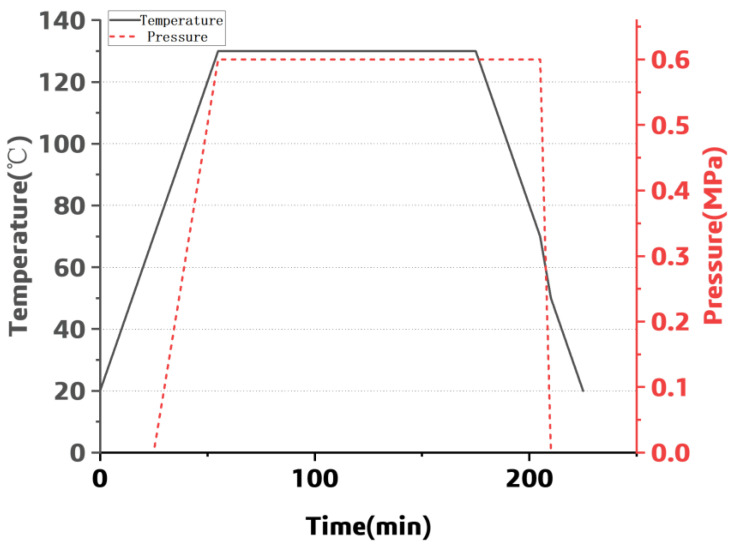
The parameters during the curing process.

**Figure 14 polymers-15-02758-f014:**
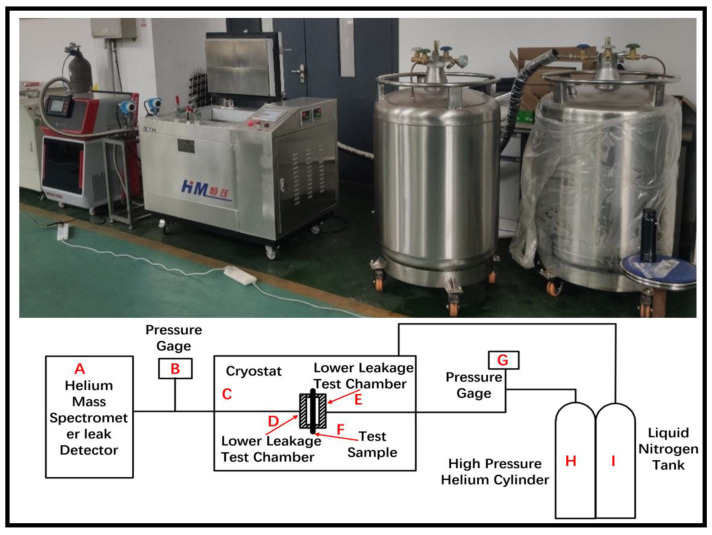
Low-temperature leak testing system diagram.

**Figure 15 polymers-15-02758-f015:**
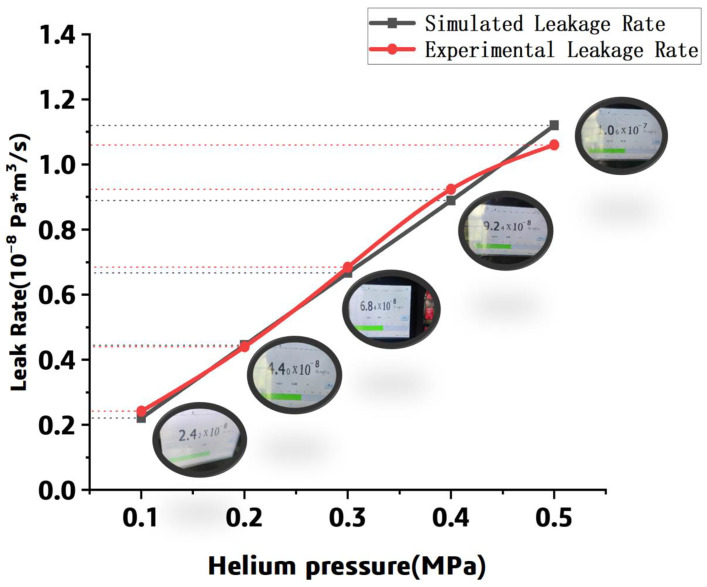
The leak rate comparison between the prediction and the experiment.

**Table 1 polymers-15-02758-t001:** The values used for the finite element analysis.

	S (kg/m^3^)	D (m^2^/s)
Air	0.178	7.20 × 10^−5^
Carbon Fiber	0	0
Interface	1.64 × 10^−6^	7.2 × 10^−7^
Polymer	2.88 × 10^−9^	3.20 × 10^−11^

**Table 2 polymers-15-02758-t002:** The simulated leak rate and the experimental leak rate (unit: 10^−8^ kg/(s⋅m^2^)).

Pressure Load (MPa)	Simulated Leak Rate	Experimental Leak Rate	Difference Value	Error Value
**0.1**	2.21	2.42	0.21	8.68%
**0.2**	4.45	4.40	−0.05	1.14%
**0.3**	6.67	6.84	0.17	2.43%
**0.4**	8.89	9.24	0.35	3.79%
**0.5**	11.2	10.6	−0.60	5.66%

## Data Availability

The data that support the findings of this study are available on request from the corresponding author upon reasonable request.

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
