# Peer review of "Simulation and Experimental Study on the Internal Leak Behavior in Carbon Fiber Reinforced Composite Components"

_polymers, 2023, doi:10.3390/polym15132758_

Round 1

Reviewer 1 Report

The paper experimentally and numerically (by FE model) studied the influence of fiber and fiber-resin interface on permeability. The novelty is good. But, the revision is necessary.

1) I suggest to remove the first three lines in the abstract (lines 10-12).

2) In order to provide a more comprehensive literature review, the authors should cite and discuss the following relevant papers in their revised manuscript:

Experimental investigation of the effect of diameter upon low velocity impact response of glass fiber reinforced composite pipes. Composite Structures2021, 275, p.114428.

SCFs in tubular X-joints retrofitted with FRP under out-of-plane bending moment. Marine Structures, 79, 2021;p.103010.

3) please increase the quality of numbers in Fig. 3b.

4) Fig. 5 needs more discussion in the text.

5)  Please add the sensitive analyze on the mesh size in FE modeling.

 6) please more discus in validation of the FE model.

7) How could fibers considerably decrease the stress? Also, how was the direction of the layers?

8) What does it add to the subject area compared with other published
material?
9) There should be referenced in the equations, figures, and tables if they are taken from somewhere else.
10) All formulas should be checked, and Italic forms in formulas should be followed in the text.

Reviewer 2 Report

1. To review the technical editing part and for al the graps you must to improve the quality and the visibility of the axis details.

2. You specified the type of finite element used (DC3D8), but you do not specify the size of the elements and how many finite elements resulted on the model after discretization.

Is not necessary to improve the English language.

Reviewer 3 Report

Dear Authors,

Gas leak in composites tanks is a very important issue in several industrial sectors, and the availability of a reliable model is fundamental.

I kindly ask the authors to consider the following observations before the publication of your work:

-       Please check the equation 1; in my opinion the diffusion of the gas should be correlated to concentration gradient, temperature and pressure; but I do not see any terms in equation 1 related to the pressure. Your equation 2 confirms that pressure should be present in equation 1.

-         Please explain more clearly the settings and calculation for the homogenized model

-        Please consider the possibility to report the pressure load value for real industrial applications (for example in aerospace as you mentioned in your introduction) in order to evaluate how representative are the testing conditions of your experiments.

It should be also interesting in future works to verify the model with the diffusion of other relevant gasses used in several industrial applications that can benefit from composites storage solutions.

Thank you very much for your work.

Round 2

Reviewer 1 Report

ok